# Influence of Metabolic, Transporter, and Pathogenic Genes on Pharmacogenetics and DNA Methylation in Neurological Disorders

**DOI:** 10.3390/biology12091156

**Published:** 2023-08-22

**Authors:** Olaia Martínez-Iglesias, Vinogran Naidoo, Iván Carrera, Juan Carlos Carril, Natalia Cacabelos, Ramón Cacabelos

**Affiliations:** EuroEspes Biomedical Research Center, International Center of Neuroscience and Genomic Medicine, 15165 Bergondo, Corunna, Spain; neurociencias@euroespes.com (V.N.); biotecnologiasalud@ebiotec.com (I.C.); genomica@euroespes.com (J.C.C.); serviciodocumentacion@euroespes.com (N.C.); rcacabelos@euroespes.com (R.C.)

**Keywords:** epigenetics, DNA methylation, pharmacogenetics, neurological disorders

## Abstract

**Simple Summary:**

The prevalence of neurodegenerative disorders has increased, yet reliable, pre-symptomatic biomarkers for early detection and subsequent preventive interventions remain elusive. The aim of our study was to understand the interplay between DNA methylation, a crucial biological process that modifies DNA activity, genetic composition, and the impact of drugs on neurodegenerative diseases. We distinguished two patient groups based on their DNA methylation levels and found unique drug-response patterns and gene-interaction dynamics within each group. These variations in gene interactions produced differential treatment responses that correlated with temporal changes in DNA methylation. Our findings highlight the critical role of specific genes, including *CYP1A2*, *CYP2E1*, *CHAT*, and various transporter genes, in shaping DNA methylation and treatment outcomes. Moreover, gene variants such as *APOE* and *NBEA* influenced disease progression and treatment response. The complexity of these interactions was further underscored by the varied responses of other genes such as the drug metabolizing genes *CYP4F2*, *NAT2*, and *COMT*. Our study highlights the potential of precision medicine in managing neurodegenerative disorders by tailoring treatments to individual genetic and epigenetic profiles. By deepening our understanding of these interactions, we can potentially improve treatment efficacy, thereby reducing the societal burden of neurodegenerative diseases.

**Abstract:**

Pharmacogenetics and DNA methylation influence therapeutic outcomes and provide insights into potential therapeutic targets for brain-related disorders. To understand the effect of genetic polymorphisms on drug response and disease risk, we analyzed the relationship between global DNA methylation, drug-metabolizing enzymes, transport genes, and pathogenic gene phenotypes in serum samples from two groups of patients: Group A, which showed increased 5-methylcytosine (5mC) levels during clinical follow-up, and Group B, which exhibited no discernible change in 5mC levels. We identified specific SNPs in several metabolizing genes, including *CYP1A2*, *CYP2C9*, *CYP4F2*, *GSTP1*, and *NAT2*, that were associated with differential drug responses. Specific SNPs in CYP had a significant impact on enzyme activity, leading to changes in phenotypic distribution between the two patient groups. Group B, which contained a lower frequency of normal metabolizers and a higher frequency of ultra-rapid metabolizers compared to patients in Group A, did not show an improvement in 5mC levels during follow-up. Furthermore, there were significant differences in phenotype distribution between patient Groups A and B for several SNPs associated with transporter genes (*ABCB1*, *ABCC2*, *SLC2A9*, *SLC39A8*, and *SLCO1B1*) and pathogenic genes (*APOE*, *NBEA*, and *PTGS2*). These findings appear to suggest that the interplay between pharmacogenomics and DNA methylation has important implications for improving treatment outcomes in patients with brain-related disorders.

## 1. Introduction

Neurological disorders (NDs) are increasingly prevalent and represent the leading cause of disability-adjusted life years [1]. They account for over 6% of the global disease burden [2], are the second most common cause of death worldwide with nine million deaths annually, and are exacerbated by severe health disparities. For example, approximately 80% of the 50 million people with epilepsy reside in low- and middle-income countries [3]. NDs can be classified into six major categories that include age-related neurodegenerative disorders (e.g., Alzheimer’s disease, Huntington’s disease, and Parkinson’s disease), mental disorders (e.g., depression, psychosis), neurotoxic disorders (e.g., alcoholism), cerebrovascular disorders (e.g., stroke), neurodevelopmental disorders (e.g., autism), and other complex disorders (e.g., epilepsy). The pathogenesis of central nervous system (CNS) disorders is complex and involves a combination of genetic and environmental factors.

There is currently a lack of pre-symptomatic biomarkers to facilitate the early detection of and intervention for NDs. Treating NDs often involves prolonged symptomatic management, carries significant costs, and has an increased risk of adverse drug reactions (ADRs). Moreover, developing effective treatments requires a comprehensive approach that considers each disorder’s unique characteristics as well as drug–drug interactions (DDIs) and the risk of ADRs in polypharmacy. To this, most patients with NDs require multifactorial treatment. Addressing the complexities of NDs, such as Alzheimer’s disease (AD), necessitates multifactorial treatments, tailored to the needs of each individual. Such treatments span a wide array of pharmacological approaches, from neuroprotective and anti-dementia drugs to medications targeting concurrent pathologies, neuropsychiatric disorders, and drugs to treat metabolic deficits. Here, the application of pharmacogenetic procedures can be particularly beneficial [4]. Over 90% of AD patients require these multifactorial treatments, which often involve treating coexisting conditions such as hypertension (>25%), obesity (>70%), type 2 diabetes mellitus (>25%), hypercholesterolemia (40%), hypertriglyceridemia (20%), metabolic syndrome (20%), hepatobiliary disorder (15%), endocrine/metabolic disorders (>20%), cardiovascular disorder (40%), cerebrovascular disorder (60–90%), neuropsychiatric disorders (60–90%), and cancer (10%) [4]. However, this strategy can increase the risk of ADRs and DDIs. As a result, ADRs and DDIs have emerged as significant global health issues that affect individuals undergoing medical treatments [5,6] and rank among the top ten leading causes of death and illness in developed countries [6]. In the United States alone, approximately half a million ADRs are reported annually, with associated direct costs that exceed USD 150 billion per year [7]. Furthermore, ADRs increase hospital admissions, lengthen hospital stays, and raise mortality rates and healthcare costs; they may also lead to drug withdrawal from the market [8]. A focus on precision medicine could improve patient outcomes and reduce the burden of NDs on individuals, families, and society.

Pharmacogenomics, the study of the influence of genetic variability on drug response and toxicity, is a vital component of precision medicine. By providing insight into how patients will respond to specific therapies, pharmacogenomics guides prescription and drug dose decisions. This helps reduce the risk, occurrence and severity of ADRs while optimizing drug efficacy [9]. Over 50% of drugs currently have a known pharmacogenomic profile that can be used to optimize efficacy and prevent adverse effects [6]. The pharmacogenomic machinery integrates drug and gene interactions through two pathways: the pharmacogenomic–pharmacokinetic pathway, which occurs during the absorption, distribution, metabolism and excretion (ADME) processes, and the pharmacogenomic–pharmacodynamic pathway, which occurs at the drug-target level [10]. Proper functioning of these pathways is essential for drug efficacy, and deficiencies or dysfunctions can cause ADRs or toxicity [11].

Pharmacogenetics accounts for approximately 80% of variability in drug safety and efficacy [9]. Rare variants make up 50% of the functional variability reported in more than 140 clinically relevant pharmagenes. Over 400 genes, including their encoded enzymes/proteins, influence drug efficacy and safety, and approximately 240 pharmagenes are associated with ADRs [12]. Pharmacogenetic outcome is influenced by various classes of genes, including pathogenic, mechanistic, metabolic, transporter and pleiotropic genes that comprise the pharmacogenetic machinery. These genes are regulated by epigenetic factors such as DNA methylation, chromatin/histone modifications, and miRNAs [4,13]. The enzymes and transporters that are integral to metabolic pathways exhibit a considerable degree of polymorphism. The presence of polymorphisms in the genes that encode these molecular components cause significant variability in interindividual drug responses [13,14]. Cytochrome P450 oxidases (CYPs) play a crucial role in regulating drug efficacy and toxicity. CYP1A2, CYP3A4/5, CYP2C9, CYP2C19 and CYP2D6 are among the most important CYPs involved in drug metabolism [15].

Genomics and epigenetics play a crucial role in the development and progression of neurodegenerative diseases. Epigenetics refers to hereditary changes in phenotype or gene expression that result from chromatin-based mechanisms, which do not involve alterations in the DNA sequence. Both biological and environmental factors modulate epigenetic modifications, which consequently impact gene expression and phenotype [16]. The accumulation of a variety of epigenetic modifications during the lifespan may lead to neurodegenerative diseases [17,18]. DNA methylation, the most extensively studied epigenetic mark, is a reversible mechanism catalyzed by DNA methyltransferases (DNMTs) that transfer a methyl group from the cofactor SAM (S-adenosyl-l-methionine) to the C5 position of cytosine found in CpG dinucleotides [19]; this results in the conversion of cytosines to 5-methylcytosines (5mC). DNA methylation changes the stability and accessibility of DNA, which regulates gene expression [18]. It is most commonly associated with gene silencing [20], which attracts additional silencing elements, such as methyl-CpG-binding proteins [21]. DNMT proteins are classified into three families: DNMT1, DNMT2, and DNMT3, all of which are expressed in neurons [22]. DNMT1 preserves methylation patterns after cell division, ensuring the inheritance of these methylation marks [23]. DNMT3a and DNMT3b play essential roles in the process of de novo methylation [24]. Reduced global DNA methylation levels identified in blood samples of individuals with NDs could potentially function as a diagnostic biomarker [25,26,27,28]. Epigenetic-based therapies may potentially restore 5mC and other epigenetic modifications, providing a novel approach that could delay or reduce disease progression and improve the quality of life of patients.

Our recent study identified two categories of patients: Group A, where 5mC levels were found to be higher during the follow-up compared to their initial visit, and Group B, where patients exhibited lower or similar 5mC levels during the follow-up as compared to their initial visit [29]. As pharmacogenetics can account for over 80% of the variability in drug pharmacodynamics and pharmacokinetics, we conducted a retrospective study to investigate the influence of genetic factors on global DNA methylation levels in patients with NDs. More specifically, we investigated the influence of metabolic (*CYP1A1*, *CYP1A2*, *CYP1B1*, *CYP2A6*, *CYP2B6*, *CYP2C9*, *CYP2C19*, *CYP2D6*, *CYP2E1*, *CYP3A4*, *CYP3A5*, *CYP4F2*, *CES1*, *CHAT*, *COMT*, *GSTM1*, *GSTP1*, *GSTT1*, *NAT2*, *SOD2*, *TPMT*, *UGT1A1*), transporter (*ABCB1*, *ABCC2*, *ABCG2*, *SLC2A2*, *SLC2A9*, *SLC6A2*, *SLC6A3*, *SLC6A4*, *SLC39A8*, *SLCO1B1*), and pathogenic (*NBEA*, *PTGS2*, *APOE*) gene variants on global DNA methylation with regard to the therapeutic outcome during the initial and follow-up clinical assessments in both groups of patients.

Our aim in the current study was to provide new insights into the complex interplay between genetic and epigenetic determinants in modulating temporal DNA methylation patterns in patients with NDs to help us better understand disease susceptibility and treatment outcomes and develop precision medicine strategies.

## 2. Materials and Methods

### 2.1. Subjects

In this retrospective study, a total of 98 patients ranging in age from 42 to 84 years old (mean age: 56 ± 1.61 years; 53 males and 45 females) were recruited via the CIBE patient database at EuroEspes International Center of Neuroscience and Genomic Medicine (C000925, 21 October 2013, EuroEspes Biomedical Research Center, Corunna, Spain). The study adhered to the Helsinki Declaration and Spanish law (Organic Law on Biomedical Research, 14 July 2007) and received approval from the Ethics/Research Committee of the EuroEspes Biomedical Research Center (Epibiomarkers EE0620). Blood samples were drawn from participants on their initial and subsequent visit to the Center. Informed consent was obtained from all subjects and/or legal caregivers. Patients were diagnosed following widely-accepted global criteria subsequent to comprehensive clinical and genetic examinations. The clinical procedure included a genomic analysis of single-nucleotide polymorphisms (SNPs) linked to Parkinson’s disease (PD), AD, or vascular risk factors, along with psychological assessments, brain mapping, and neuroimaging.

Patients were classified into two groups based on their 5mC levels. Group A consisted of patients whose 5mC levels were elevated during the follow-up visit compared to the initial consultation. Group B constituted those subjects whose 5mC levels remained consistent or decreased during the follow-up. This stratification process was performed manually, accounting only for variations in 5mC levels exceeding 0.2% for differentiating patients into Groups A and B.

### 2.2. Sample Collection and Analysis

Peripheral venous blood samples were obtained from fasting individuals in a supine position. The samples were collected into EDTA-coated tubes, followed by centrifugation at 3000 rpm for 10 min at 4 °C to isolate the buffy coat. Subsequently, the buffy coat was stored at −40 °C until DNA extraction. For serum collection, blood was allowed to clot for 30 min at room temperature before being centrifuged at 1500 rpm for 10 min at 4 °C. The resulting supernatant (serum) was removed and stored at −80 °C.

### 2.3. DNA Extraction

Peripheral blood lymphocyte DNA was isolated using the QIAcube robotic workstation, together with the QIAamp DNA Mini Kit (Qiagen, Venlo, The Netherlands), following the manufacturer’s recommended protocol. The purity and concentration of the extracted DNA were determined using a microplate spectrophotometer (Epoch, BioTek Instruments, Winooski, VT, USA). Only DNA samples exhibiting 260/280 and 260/230 ratios greater than 1.8 were included in this study.

### 2.4. Quantification of Global DNA Methylation (5mC)

The quantification of global 5mC levels was performed colorimetrically with the MethylFlash Methylated DNA Quantification Kit (EpiGentek, Farmingdale, NY, USA) with 50 ng of DNA per sample according to the manufacturer’s instructions. Absorbance was measured at 450 nm using a microplate reader. A standard curve was constructed using linear regression (Microsoft Excel) to determine the absolute quantity of methylated DNA. The amount and percent of 5mC were calculated using the following formulas:5mC (ng) = (Sample OD − Blank OD)/(Slope × 2)
5mC (%) = 5mC (ng)/sample DNA (ng) × 100 

### 2.5. Genotyping

The aim of this study was to examine the impact of polymorphisms in ND-related genes on their mRNA expression levels. For genotyping SNPs, we used qPCR with TaqMan assays and used the StepOnePlus Real-Time PCR System (Life Technology, Carlsbad, CA, USA) with TaqMan OpenArray DNA microchips for the QuantStudioTM 12K Flex Real-Time PCR System. The resulting genotyping data were analyzed using Genotyper software (Thermo Fisher Scientific, Waltham, MA, USA).

### 2.6. Statistical Analysis

Comparisons between groups was based on the global distribution of geno-phenotypes and was performed using chi-square tests without Yates correction (GraphPad Prism, San Diego, CA, USA). A *p*-value less than 0.05 was considered statistically significant.

## 3. Results

### 3.1. Genetic Variability in Drug Metabolizing Enzymes Influences 5mC Levels

We first investigated the influence of genetic variability in drug metabolizing enzymes, which are involved in regulating drug efficacy and toxicity [30,31], on the levels of 5-methylcytosine (5mC). Here, we analyzed the frequency of different phenotypes of cytochrome P450 (CYP) enzymes in two distinct groups (A and B) of patients based on their 5mC levels during their initial and follow-up visits. Phase I metabolism, which converts lipophilic drugs into more polar molecules by oxidation, reduction, or hydrolysis reactions, plays a key role in drug elimination [32], and over 2000 mutations have been described in *CYP* genes, some of which have a substantial impact on CYP activity [33]. More specifically, we investigated the impact of SNPs on the activity of CYP enzymes, including CYP1A1, CYP1A2, CYP1B1, CYP2A6, CYP2B6, CYP2C9, CYP2C19, CYP2D6, CYP3A4, CYP3A5, and CYP4F2.

Based on previous findings demonstrating the influence of genetic variability in drug metabolizing enzymes on 5mC levels, we further analyzed the frequency of different CYP phenotypes in two distinct patient groups: those whose 5mC levels increased during follow-up (Group A) and those whose levels decreased or remained similar to the initial visit (Group B) [29]. For *CYP1A1* rs1378942, Group A showed a distribution of 20% abnormal metabolizers, 59% deficient metabolizers, and 21% normal metabolizers, while Group B had 28% abnormal, 59% deficient, and 21% normal metabolizer phenotypes (Figure 1A). For CYP1A2, the percentage of normal metabolizers decreased from 75% to 40% in Group B compared to Group A, and the percentage of ultra-rapid metabolizers increased from 25% to 60% (Figure 1B). The SNPs analyzed were rs2069514, rs35694136, and rs762551. In the case of *CYP1B1* rs1056836 phenotypes, a rapid metabolizer was detected in 92% (Group A) and 96% (Group B) of cases, while the normal metabolizer was observed in 8% and 4% of cases, respectively (Figure 1C). For rs28399433 CYP2A6 phenotype analysis, a normal metabolizer incidence of 92% in Group A and 96% in Group B was observed (Figure 1D). The intermediate metabolizer phenotype showed an incidence of 8% in Group A and 4% in Group B (Figure 1D). No changes were detected in rs3745274 CYP2B6 phenotypes between Group A and B, with 68% and 65% normal metabolizers, 29% and 31% intermediate metabolizers, and 3% and 4% poor metabolizers, respectively (Figure 1E).

For CYP2C9, the percentage of normal metabolizers increased from 68% to 88% in Group B compared to Group A, and the percentage of intermediate metabolizers decreased from 29% to 12% (Figure 1F). The poor metabolizer phenotype was only detected in Group A (3%) (Figure 1F). The SNPs analyzed were rs1057910, rs1799853, rs28371685, rs28371686, rs7900194, and rs9332131. In the case of CYP2C19, the percentage of normal metabolizers increased from 57% to 64% in Group B compared to Group A, and the percentage of intermediate metabolizers increased from 14% to 24% (Figure 1G). The percentage of ultra-rapid metabolizers decreased from 29% to 12% (Figure 1G).

We next examined the effects of specific SNPs on CYP2D6, CYP3A4, CYP3A5, and CYP4F2 phenotypes in patients in Groups A and B. Analysis of the rs12248560 and rs4244285 SNPs in CYP2D6 showed that the percentage of ultra-rapid metabolizers decreased from 29% to 12% in Group B (Figure 1G). After analyzing the SNPs rs28371725, rs35742686, rs3892097, and rs5030655, there were no significant changes in the phenotypic distribution between Groups A and B in CYP2D6, with 48% and 54% of normal metabolizers, 37% and 32% of intermediate metabolizers, and 15% and 14% of ultra-rapid metabolizers, respectively (Figure 1H). Similarly, there were no significant changes in the phenotypic distribution of the *CYP3A4* SNPs rs2242480 and rs35599367 between Group A and B (66% and 76% normal metabolizers, 23% and 16% intermediate metabolizers, and 11% and 8% poor metabolizers, respectively) (Figure 1J). In contrast, the rs776746 SNP in *CYP3A5* showed an increase in the percentage of normal metabolizers from 69.7% to 84% in Group B, while the percentage of rapid metabolizers decreased from 30.3% to 12% (Figure 1K). Furthermore, only Group B had intermediate metabolizers (4%) (Figure 1K). Analysis of the rs2108622 SNP in *CYP4F2* showed an increase in the percentage of normal metabolizers from 51% to 84% in Group B, a decrease in intermediate metabolizers from 37% to 12%, and a decrease in poor metabolizers from 12% to 4% (Figure 1L).

After examining the effects of specific SNPs on CYP phenotypes in patient Groups A and B, we proceeded to analyze SNPs that are associated with phase II metabolism. This process involves conjugation reactions that increase drug solubility in water [32]. Our analysis of the rs71647871 SNP in the *CES* gene revealed that all cases in Group A were poor metabolizers. In contrast, 96% of cases in Group B were poor metabolizers and 4% were intermediate metabolizers (Figure 2A). For the rs2177369 SNP in the *CHAT* gene, there was a decrease in normal metabolizers from 35% to 24% and an increase in intermediate metabolizers from 27% to 48% in Group B compared to Group A. The percentage of poor metabolizers decreased from 38% to 28% (Figure 2B). When analyzing the rs4680 SNP in the *COMT* gene, there was a decrease in normal metabolizers from 29% to 20%, a decrease in poor metabolizers from 24% to 12%, and an increase in intermediate metabolizers from 47% to 68% (Figure 2C). For the Indel CNV in the *GSTM1* gene, which is characterized by a complete deletion (GSTM1*0), there was an increase in normal metabolizers from 63% to 72% in Group B, a decrease in poor metabolizers from 29% to 20%, and no significant change for intermediate metabolizers between Groups A and B (Figure 2D). Finally, our analysis of rs1138272 and rs1695 SNPs in *GSTP1* showed an increase of normal metabolizer phenotypes from 47% to 76% (Group B), while intermediate and poor metabolizer phenotypes decreased, respectively, from 38% to 16% and 15% to 8% (Figure 2E).

By studying multiple SNPs in different genes, we next aimed to identify additional genetic factors that influence drug metabolism and inter-individual differences in patient drug responses. Therefore, to better understand the genetic variations that contribute to drug metabolism in our patient populations, we analyzed SNPs or copy number variants (CNVs) in *GSTT1*, *NAT2*, *SOD2*, *TPMT*, and *UGT1A1*. The *GSTT1* gene encodes a member of the glutathione S-transferase (GST) family of enzymes, which play an important role in the detoxification of xenobiotics, including drugs by conjugating them with glutathione [32]. By examining the insertion and deletion (indel) variant (CNV) of the *GSTT1* gene, we assessed whether this polymorphism had an effect on the overall drug metabolism in our patient cohorts, and whether it contributed to the observed differences in metabolizer phenotypes between patients in Groups A and B. Our analysis of the Indel CNV of the *GSTT1* gene revealed an increase in the proportion of normal metabolizers from 21% to 28%, while intermediate metabolizers showed a slight increase from 50% to 56%, and poor metabolizers decreased from 29% to 16% (Figure 2F). Additionally, we analyzed seven different *NAT2* SNPs (rs1041983, rs1208m, rs1700020, rs1799930, rs1799931m, rs1801279, and rs1801280) and found that the incidence of slow acetylators decreased from 46% to 36% in Group B, whereas fast acetylators increased from 3% to 24% (Figure 2G). The incidence of intermediate acetylators decreased from 51% to 40%.

Our analysis of the rs4880 SNP in the *SOD2* gene sought to investigate its potential impact on drug metabolism in our patient cohorts. This SNP has been associated with various diseases and oxidative stress and may also influence enzyme activity [34]. Therefore, we examined its distribution in Groups A and B to further understand its relationship with drug metabolism. For the rs4880 *SOD2* SNP, we observed a similar distribution of patients with normal (32% and 28% in Groups A and B, respectively) as comparing with patients showing an abnormal activity (67% and 72%) (Figure 2H).

The *TPMT* gene encodes the thiopurine S-methyltransferase enzyme, which is responsible for the metabolism of thiopurine drugs widely used for the treatment of leukemia and autoimmune disorders [35]. Genetic polymorphisms within *TPMT* significantly impact the activity of the TPMT enzyme, causing interindividual variability in response to thiopurine treatment [36]. In the current study, to determine the distribution of *TPMT* SNPs in Groups A and B, which have differential responses to drug treatment, we analyzed four specific SNPs, rs1142345, rs1800460, rs1800462, and rs1800584. Our aim was to investigate whether genetic differences in *TPMT* contribute to the observed differences in drug response between the two groups. Analysis of the *TPMT* SNPs rs1142345, rs1800460, rs1800462, and rs1800584 showed that most patients exhibited a normal activity phenotype, with only 4% being identified as patients with abnormal activity in Group B (Figure 2I).

The *UGT1A1* gene is implicated in the development of several cancer subtypes, including colon, breast, and prostate cancer. The UGT1A1 enzyme is involved in the metabolism of various drugs, including anticancer agents [37]. The *UGT1A1* gene is significantly polymorphic, and several SNPs impact UGT1A1 enzyme activity. Certain UGT1A1 variants have been associated with a heightened risk of toxicity in response to particular drugs, including the chemotherapy drug irinotecan [38,39]. In the current study, we analyzed the distribution of *UGT1A1* SNPs in patient groups A and B to evaluate their potential role in drug metabolism and toxicity in these population cohorts. Analysis of the distribution of *UGT1A1* SNPs rs35350960, rs14124874, rs4148323, and rs887829 revealed a slight increase in the proportion of normal metabolizers from 63% (Group A) to 72% (Group B) (Figure 2J). The incidence of poor metabolizers decreased from 29% (Group A) to 20% (Group B), while the incidence of intermediate metabolizers remained unchanged in both groups (8%).

### 3.2. Modification of DNA Methylation Levels by Transport Gene Phenotypes

Our next objective was to identify genetic factors that could influence patient response to treatment by analyzing transport gene phenotypes. To achieve this, we determined the frequencies of normal, deficient, and abnormal response phenotypes for each gene in patient groups A and B. By examining the genetic profiles of these patients, we aimed to elucidate the mechanisms underlying treatment outcomes and provide insights into precision medicine. Specifically, we investigated whether specific SNPs could modify DNA methylation levels and alter phenotypic responses to treatment. We analyzed phenotypic changes in response to treatment for several SNPs in the *ABCB1*, *ABCC2*, *ABCG2*, *SLC2A2*, *SLC2A9*, *SLC6A2*, and *SLC6A3* genes.

Analysis of the *ABCB1* SNPs rs1128503, rs1032582.01, rs2032582.02 and rs1045642 revealed a significant decrease in abnormal response phenotype from 60% to 32% in Group B (Figure 3A). Normal and deficient response phenotypes increased from 23% to 40%, and from 17% to 28%, respectively (Figure 3A). *ABCC2* SNPs rs717620, rs2273697, rs17222723 and rs3740066 showed an increase in the normal response phenotype from 26% to 36% in Group B, whereas the deficient response phenotype increased from 53% to 60% (Figure 3B). A significant decrease from 21% to 4% in the abnormal response phenotype was observed in Group B (Figure 3B).

*ABCG2* SNP rs2231142 showed a similar frequency of the normal response phenotype in Groups A (85%) and B (84%) (Figure 3C). The principal difference between the two groups was the presence of an abnormal response phenotype (3%) in Group A (Figure 3C). Analysis of the rs5400 SNP in the *SLC2A2* gene showed only minor differences in phenotype distribution between the two groups (Figure 3D). In contrast, normal response phenotypes were found in 9% of Group A patients and in 12% of Group B patients, while deficient response phenotypes were observed in 26% and 36% of Group A and B patients, respectively (Figure 3D). In Groups A and B, abnormal responses were observed in 65% and 52% of patients respectively. The normal response phenotypes were observed in 9% of Group A patients and in 12% of Group B patients. Deficient response phenotypes were observed in 26% of Group A patients and 36% of Group B patients (Figure 3D).

Analysis of the *SLC2A9* SNP rs16890979 revealed a significant reduction in the deficit response phenotype in Group B, decreasing from 41% in Group A to 28% in Group B (Figure 3E). The normal response phenotype increased from 56% (Group A) to 64% (Group B), while the altered response increased slightly from 3% in Group A to 8% in Group B (Figure 3E). Analysis of the rs5569 SNP in the *SLC6A2* gene indicated minor differences in the frequency of the various phenotypes (Figure 3F). Specifically, the normal response phenotype increased from 9% in Group A to 12% in Group B, while the deficient response phenotype increased from 26% in Group A to 36% in Group B. The abnormal response decreased from 65% (Group A) to 52% (Group B) (Figure 3F). Analysis of the *SLC6A3* SNP rs460000 revealed no change in phenotype distribution between Groups A and B (Figure 3G). The frequency of normal response phenotypes was 66% and 64% for Groups A and B, respectively. For the deficient response phenotype, it was 31% and 32%, and for the abnormal response phenotype, it was 3% and 4%, for Groups A and B, respectively (Figure 3G). The *SLC6A4* SNP rs2020936 showed an increase in normal response phenotypes from 50% in Group A to 64% in Group B (Figure 3H). Both abnormal and deficient phenotypes decreased from 18% to 12% and from 32% to 24%, in Groups A and B, respectively (Figure 3H). The SLC39A8 SNP rs13107325 showed a marked reduction in the frequency of normal response phenotypes from 85% in Group A to 59% in Group B (Figure 3I). Abnormal response phenotypes were observed in 15% and 27% of patients in Groups A and B, respectively, whereas the deficient response phenotype was only observed in 14% of Group B subjects (Figure 3I). Analysis of the *SLCO1B1* SNPs rs2306283, rs4149015, and rs1419056 showed a strong increase in the frequency of the normal response phenotype from 49% in Group A to 92% in Group B, with only minor changes in the deficient response phenotype observed in 11% and 8% of patients in Groups A and B, respectively (Figure 3J). Interestingly, an abnormal response phenotype was only present in Group A, with a frequency of 40% (Figure 3J).

### 3.3. Pathological Gene Phenotypes Alter DNA Methylation Levels

Altered DNA methylation levels caused by pathological gene phenotypes have been implicated in numerous diseases, including AD, for which apolipoprotein E4 (APOE4) is a major risk factor [40]. The E2 allele of the *APOE* gene is protective against AD [41]. To gain insight into the role of genetic factors in modifying DNA methylation levels and their potential impact on disease risk and drug response, we next analyzed the frequency of different genotypes in Groups A and B and found significant differences in the distribution of *APOE*, neurobeachin (*NBEA*), and prostaglandin-endoperoxide synthase 2 (*PTGS2*) genotypes.

The frequency of the *APOE 3.3* genotype decreased from 73% in Group A to 56% in Group B (Figure 4). Furthermore, the frequency of the pathological *APOE 4.4* genotype increased from 2.7% in Group A to 8% in Group B. The *APOE 2.4* genotype was only present in Group B, at a frequency of 4%.

Neurobeachin (NBEA) is a kinase-anchoring protein that contributes to synapse maturation and development of the nervous system [42]. The *NBEA* SNP rs1779800 is associated with differential therapeutic responses to acetylcholinesterase inhibitors (AChEIs) [43]. In the current study, there was a substantial decrease in the normal response phenotype from 64% (Group A) to 35% (Group B), and a corresponding increase in the deficient response phenotype from 27% (Group A) to 55% (Group B) (Figure 5). However, the abnormal response phenotype showed similar values in both groups A and B (9% and 10%, respectively).

The emergence of behavioral disorders in AD may be associated with certain genetic polymorphisms, metabolic dysfunctions, and cerebrovascular risk factors. The symptomatology of behavioral disorders in different forms of dementia is not uniform, but depression and apathy are common manifestations that often necessitate drug intervention. The reported frequency of depression in AD varies widely, ranging from 5% to over 40% [44]. Nonetheless, after apathy, depression is identified consistently as the second most prevalent psychiatric symptom in AD, and its incidence increases in advanced stages of the disease [43]. The *PTGS2* SNP rs5275 has been implicated in the development of major depressive disorder (MDD) [45,46]. Our analysis of this SNP revealed an increase in the normal response phenotype from 46% in Group A to 67% in Group B, accompanied by a substantial decrease in the frequency of the abnormal response phenotype from 15% (Group A) to 1% (Group B) (Figure 6). These data provide insight into the genetic factors that influence patient response to treatment and have important implications for the development of precision medicine strategies for NDs.

## 4. Discussion

DNA methylation has emerged as a promising biomarker for brain-related disorders as it provides insights into gene expression patterns and potential therapeutic targets [25,26]. In a recent study, we examined the correlation between hypovitaminosis, psychometric parameters, DNA methylation, and NDs, and found two distinct patient categories based on changes in 5mC levels during clinical follow-up: Group A, which showed increased 5mC levels during the follow-up, and Group B, which exhibited no discernible change in 5mC levels [29]. In the current study, we investigated, in those two groups of patients, how different pharmacogenetic patient phenotypes, encoded by metabolizing, transporter and pathogenic genes, affect DNA methylation during clinical follow-up. The proportion of *CYP1A2* and *CYP2E1* normal metabolizers decreased in Group B, indicating the importance of *CYP1A2* in responding to changes in DNA methylation. In CHAT, a similar pattern emerged, with normal metabolizers decreasing from 35% to 24% and intermediate metabolizers increasing from 27% to 48% in this group. A substantial proportion of patients in Group B showed deficient responses to ABCB1, SLC2A2, SLC6A2, and SLC39A8. Moreover, an increase in the frequency of the *APOE4* allele was observed in Group B, implying a greater likelihood of adverse prognosis and pathogenicity in this group. Consistent with this, an increase in deficient response geno-phenotypes from 27% to 55% was observed for the *NBEA* gene. Our major finding was that variations in genotype–phenotype interactions cause different responses to treatments and are associated with differential changes in 5mC levels over time.

While a positive correlation has been established between global DNA methylation levels and age [47], our research group previously found a significant correlation between age and 5mC levels only in patients with PD [25]. Recently, we observed no correlation between age and 5mC levels in patients from Group A [29]. Despite the absence of a correlation between global DNA methylation and sex in our previous study [25], it is important to note that there is a nearly equal distribution of genders, with 53 males and 45 females.

Our first analysis examined the impact of genetic variability in drug metabolizing enzymes on 5mC levels. We focused on different phenotypes of CYP enzymes in the two groups of patients and found that specific SNPs had a significant impact on CYP enzyme activity, leading to changes in phenotypic distribution between the two groups of patients. In particular, there were no significant changes in the phenotypic distribution for some SNPs, while others showed a shift in the percentage of different metabolizing phenotypes.

To identify additional genetic factors that influence drug metabolism and inter-individual differences in patient drug responses, we next analyzed the effects of specific SNPs and CNVs on CYP and phase II metabolism phenotypes in patient groups A and B. We found that genetic polymorphisms in *CES*, *CHAT*, *COMT*, *GSTM1*, *GSTP1*, *GSTT1*, *NAT2*, *SOD2*, *TPMT*, and *UGT1A1* contributed to the observed differences in drug response between the two groups. Furthermore, there was an increase in normal metabolizers in some genes, whereas others showed a decrease in poor metabolizers and an increase in intermediate metabolizers.

By analyzing transport gene phenotypes, we then assessed whether specific SNPs could modify DNA methylation levels and alter phenotypic responses to treatment. Specifically, we examined the frequencies of normal, deficient, and abnormal response phenotypes for each gene in patient groups A and B. There were significant differences in phenotype distribution between the two patient groups for several SNPs associated with *ABCB1*, *ABCC2*, *ABCG2*, *SLC2A2*, *SLC2A9*, *SLC6A2*, *SLC6A3*, *SLC6A4*, *SLC39A8*, and *SLCO1B1*. *SLCO1B1* SNPs had a strong effect on patient response to treatment, with a marked increase in the frequency of the normal response phenotype in Group B compared to Group A.

Finally, to understand the role of pathogenic gene phenotypes in modifying DNA methylation levels and their impact on disease risk and drug response, we then analyzed the distribution of different genotypes in the two groups of patients and found substantial differences in the frequencies of *APOE*, *NBEA*, and *PTGS2* genotypes.

CYP enzymes constitute a diverse group of drug-metabolizing enzymes that regulate the metabolism of the majority of xenobiotic substances [48]. CYPs play a crucial role in about 80% of oxidative metabolism and approximately 50% of drug elimination for drugs currently in use [6]. These enzymes are linked to different gene variants that give rise to different phenotypes, including normal metabolizers (NM), intermediate metabolizers (IM), poor metabolizers (PM), and ultra-rapid metabolizers (UM), which determine drug efficacy and toxicity [6,49,50]. CYP1A2, for example, metabolizes various drugs, including phenacetin, caffeine, clozapine, tacrine, propranolol, and mexiletine [51]. In particular, a strong correlation exists between coffee consumption and PD among slow metabolizers of caffeine homozygous for *CYP1A2* polymorphisms [52]. In patients with schizophrenia, clozapine dosage is influenced by CYP1A2 activity, and tacrine metabolism is primarily dependent on the activity of CYP1A2 and CYP3A4 enzymes [53,54]. Tacrine, a reversible cholinesterase inhibitor [55], is a major substrate of CYP1A2 and CYP3A4 [56]. Propanolol is commonly prescribed for migraine prophylaxis and anxiety treatment [57,58]. Our findings suggest that Group B, in which patients did not show an improvement in 5mC levels during follow-up, contained a lower frequency of subjects that were normal metabolizers (40%, compared to 75% in Group A) and a higher frequency of patients that were ultra-rapid metabolizers. CYP1A2 is important for the dosing of several antipsychotics. Ultra-rapid metabolizers are resistant to clozapine treatment, and improved outcomes are achieved by co-administration of the CYP1A2 inhibitor fluvoxamine and by increasing clozapine dosage [59]. The higher frequency of ultra-rapid metabolizers in Group B patients may therefore explain the poor response to treatment and lack of improvement in 5mC levels during the clinical follow-up.

CYP2E1 is involved in the metabolism of fatty acids, which are abundant in the brain [60] and in the biotransformation of exogenous compounds [61]. These compounds include ethanol, nicotine, acetaminophen, acetone, aspartame, chloroform, chlorzoxazone, tetrachloride, and antiepileptic drugs such as phenobarbital. In transgenic (APP/PS1) AD mice, chlorzoxazone is neuroprotective by attenuating neuroinflammation and neurodegeneration [62]. Our results indicate that in Group B, the frequency of the normal metabolizer phenotype was lower (76% vs. 89%) and the frequency of the intermediate metabolizer phenotype was higher (24% vs. 11%) compared to Group A. Intermediate metabolizers display reduced enzymatic activity and increased side effects because of incomplete drug metabolism compared to normal metabolizers, indicating that a lower dose may be required [63]. The increase in the frequency of intermediate metabolizers in Group B subjects may contribute to the observed increase in toxicity issues and lack of improvement in 5mC levels during the patient follow-up period.

CYP4F2 is involved in the metabolism of fatty acids and vitamin E [64]. Vitamin E is associated with decreased DNMT expression [65]. Our current findings showed that Group B had a higher frequency of the normal metabolizer phenotype (82%) compared to Group A (52%). Moreover, the frequencies of intermediate and poor metabolizers were lower in Group B. This suggests that patients in Group B may be able to metabolize vitamin E more effectively than patents in Group A, which could potentially decrease DNA methylation.

Variants in the *CHAT* gene may influence the response to AChEIs [66]. In our study, the frequency of the intermediate metabolizer phenotype increased from 27% to 48% in Group B. Intermediate metabolizers have reduced enzymatic activity and increased side effects, which could explain the lack of response to treatment and the absence of improvement in 5mC levels in Group B patients. Similarly, COMT analysis showed an increase in the frequency of the intermediate metabolizer phenotype from 47% (Group A) to 68% in Group B. COMT enzyme activity is linked to various psychiatric and neurological disorders [67]. COMT is involved in Phase II metabolism and transfers a methyl group from S-adenosylmethionine (SAM) to a hydroxyl group on the catechol ring of endogenous and xenobiotic catechol substrates. During COMT-catalyzed methylation, SAM is converted to a competitive inhibitor, S-adenosylhomocysteine (SAH), resulting in a negative feedback loop [68]. Endogenous substrates of COMT include dopamine, norepinephrine, and epinephrine [67] and there is a significant association between the *COMT* rs4680 SNP and the response to antidepressant treatment [69].

Although previous studies found no significant correlations between *NAT2* genotypes and AD or PD [69,70], our present study found a higher frequency of rapid acetylators in Group B (24%) than in Group A (35%) patients. This increased frequency of rapid acetylators could contribute to the stable levels of 5mC. AtreMorine, a novel compound derived from the *Vicia faba* L. plant through a non-denaturing biotechnological methods, enhances DNA methylation in PD patients by supplying L-DOPA, a dopamine precursor widely used for treating PD [71,72]. Pharmacogenomic analysis of NAT2 phenotypes in response to AtreMorine show that the increase in DNA methylation is not statistically significant in patients with the fast acetylator phenotype [72].

Polymorphisms in drug transporter genes affect drug pharmacokinetics and ultimately drug concentration in plasma and target tissues [73]. Among these transporters, ABCB1 plays a critical role in the brain [74]. In PD, the frequency of ABCB1 phenotypes is 24.76% for low responders (LR), 32.67% for intermediate responders (IR), and 42.57% for high responders (HR), with no significant differences in response to AtreMorine among the phenotypes [75]. ABCB1 also transports 42% of antiepileptic drugs and 16% of benzodiazepines [43]. ABCB1 is a major transporter (55%) of antidepressants [76] and benzodiazepines (16%); antidepressants are substrates (25%) and inducers (3%) of ABCB1 [76]. Our findings showed a substantial decrease in the high responder phenotype in Group B, from 60% to 32%. SLC39A8 facilitates the transport of drugs for treating anxiety, panic attacks, sleep disorders, agitation, and behavioral anomalies [43]. In our study, the normal response SLCOB1 phenotype increased from 49% (Group A) to 92% in Group B patients, while only an intermediate response phenotype was observed in Group A.

ApoE plays a crucial role in the transport of lipids and cholesterol in the CNS [77]. The presence of the *APOE4* genotype significantly increases the risk of late-onset AD [77]. The severity of synucleinopathies is associated with the *APOE4* variant, independent of concomitant AD severity [78,79]. While *APOE4* is not a risk factor for PD, it increases the risk of developing dementia and cognitive decline [77]. The *APOE* genotype also affects the age of onset and severity of stroke, and individuals with the *APOE4* allele exhibit delayed recovery of verbal memory function [80] and an increased risk of developing stroke-associated dementia [81]. We and others have shown that *APOE* expression decreases in venous blood and plasma samples in AD patients, suggesting its potential as a diagnostic biomarker for the disease [81,82]. The expression of *APOE* mRNA is lower in E4 carriers than in individuals with *APOE 2.3* and -*3.3* genotypes [29]. The incidence of the *APOE4* allele was higher in Group B (20% 3.4, 8% 4.4, and 4% 2.4) than in Group A (16% 3.4 and 3% 4.4), which may explain the lack of improvement in 5mC levels during the follow-up.

In patients with AD, the *NBEA* SNP rs17798800 is a potential predictor of response to treatment with AChEIs [83]. In patients with AD and depression, the observed phenotype frequency of this SNP for AChEI responders (AR), AChEI intermediate responders (DR), and AChEI non-responders (NR) phenotypes are 8.72%, 22.56%, and 68.72%, respectively [84], which is similar to the phenotype distribution in Group A of our patient cohort. However, in the current study, we observed a different frequency distribution of the NBEA phenotype in Group B patients, with a significant decrease in the normal response frequency (10% AR, 55% DR, and 35% NR). This difference in frequency distribution suggests that the NBEA phenotype may play a role in the changes in 5mC levels observed during the follow-up period.

The present study proposes that analyzing genotypes related to metabolism, transporters, and pathological pharmacogenetics can be used to analyze the evolution of global DNA methylation levels (Table 1). Other studies have highlighted the correlation between neurodegenerative diseases and decreased levels of sirtuin activity, neurodegenerative gene expression, and global DNA methylation [25,26,81]. Furthermore, DNA methylation holds potential as a biomarker for assessing disease progression in patients with neurological disorders [29].

RNA methylation, particularly the N6-methyladenosine (m6A) modification, has emerged as a crucial regulatory mechanism in NDs, demonstrating the intricate interplay among epigenetics, RNA metabolism, and neuronal function. The m6A RNA modification has been a focal point of research in neurons derived from induced pluripotent stem cells. These cells originate from patients with amyotrophic lateral sclerosis (ALS) or frontotemporal dementia (FTD), both of which are associated with a C9orf72 repeat expansion [85]).

In ALS and FTD, a reduction in the m6A RNA modification disrupts gene expression and accelerates neurodegeneration [86]. In both diseases, the m6A modification is closely linked to TDP43 binding and autoregulation [85]. The presence of extensive RNA hypermethylation in the ALS spinal cord corresponds to methylated TDP43 substrates. Furthermore, the canonical m6A reader, YTHDF2, has been identified as a modulator of TDP43-mediated toxicity, and its knockdown has shown promising results in extending the survival of neurons carrying ALS-associated mutations. This highlights the potential for investigating the influence of pharmacogenetics on RNA methylation, providing valuable insights into the complex mechanisms underlying these diseases.

Abnormalities in m6A mRNA methylation have been linked to differentially expressed genes in AD [87]. Dysregulation of mitochondrial and cytosolic tRNA m1A methylation has also been associated with age-related disorders such as AD. METTL3-dependent RNA m6A dysregulation contributes to neurodegeneration in AD through aberrant cell cycle events, emphasizing the complexity of epitranscriptomic control in AD. This highlights the multifaceted role of RNA methylation in AD, its potential as a therapeutic target, and its contribution to our understanding of NDs.

## 5. Conclusions

Genetic polymorphisms significantly impact drug response and disease risk, as demonstrated by our examination of the relationships between global DNA methylation and drug-metabolizing enzymes, transport genes, and pathogenic gene phenotypes. We identified specific SNPs in *CYP*, *CES*, *CHAT*, *COMT*, *GSTM1*, *GSTP1*, *GSTT1*, *NAT2*, *SOD2*, *TPMT*, *UGT1A1*, *ABCB1*, *ABCC2*, *ABCG2*, *SLC2A2*, *SLC2A9*, *SLC6A2*, *SLC6A3*, *SLC6A4*, *SLC39A8*, and *SLCO1B1* genes that were associated with differential drug responses. There was also a higher frequency of ultra-rapid metabolizers in patient Group B, which may explain their poor response to treatment and lack of improvement in 5mC levels. Our study highlights the potential of DNA methylation as a biomarker for brain-related disorders and emphasizes the significance of incorporating pharmacogenomics into personalized treatment plans for brain disorders, which may improve patient treatment outcomes.

## Figures and Tables

**Figure 1 biology-12-01156-f001:**
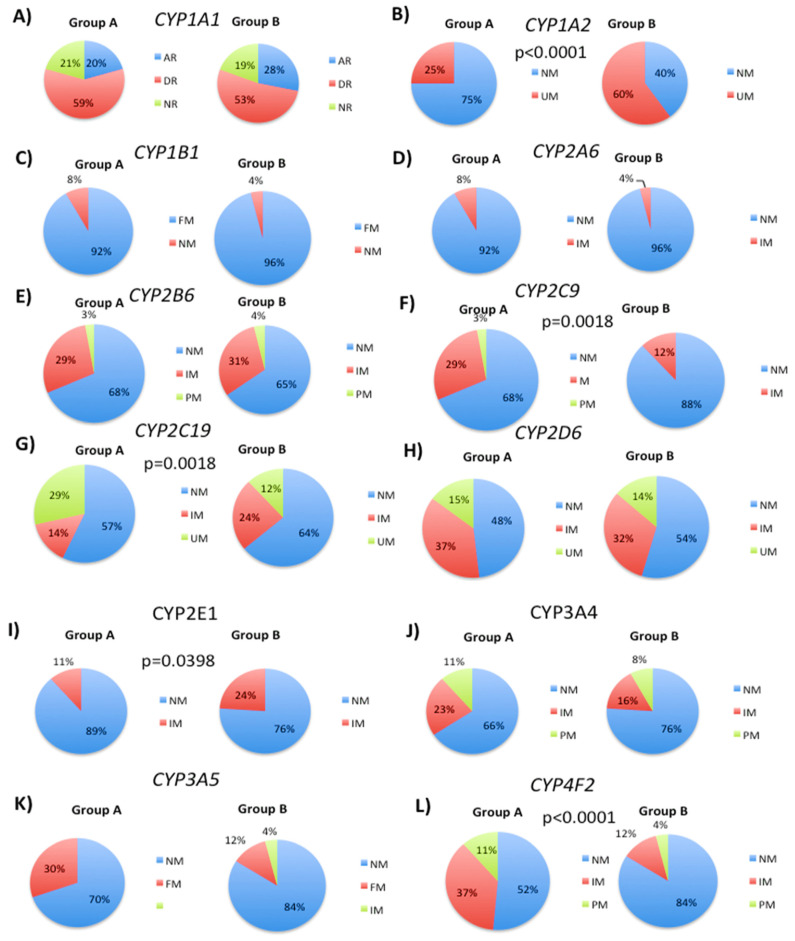
Distribution of CYP phenotypes in patient groups A and B based on changes in 5mC levels. The impact of genetic variability in drug metabolizing enzymes on 5mC levels was investigated by analyzing the frequency of different CYP phenotypes in two patient groups: those whose 5mC levels increased during follow-up (Group A) and those whose levels decreased or remained similar to the initial visit (Group B). The distribution of CYP phenotypes was analyzed for each SNP. (**A**) *CYP1A1* rs1378942, (**B**) *CYP1A2* rs2069514, rs35694136, and rs762551, (**C**) *CYP1B1* rs1056836, (**D**) *CYP2A6* rs28399433, (**E**) *CYP2B6* rs3745274, (**F**) *CYP2C9* rs1057910, rs1799853, rs28371685, rs28371686, rs7900194, and rs9332131, (**G**) *CYP2C19* rs12248560 and rs4244285, (**H**) *CYP2D6* rs28371725, rs35742686, rs3892097, and rs5030655, (**I**) *CYP2E1* rs3813867 and rs6413429 (**J**) *CYP3A4* rs2242480 and rs35599367, (**K**) *CYP3A5* rs776746, and (**L**) *CYP4F2* rs2108622 SNP. Percentages of normal, intermediate, fast, deficient, and abnormal metabolizer phenotypes are indicated for each SNP. Chi-square tests were used to compare the overall distribution of CYP phenotypes between Groups A and B. The *p*-values indicate the CYP phenotypes that are significantly different between patient groups A and B.

**Figure 2 biology-12-01156-f002:**
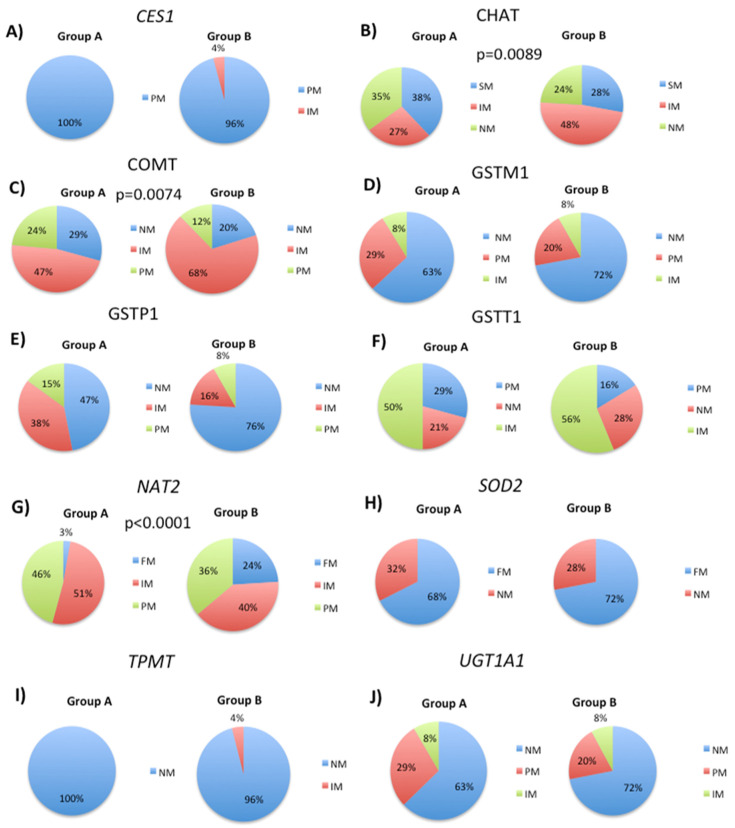
Distribution of different metabolizer phenotypes in patient groups A and B based on the analysis of several SNPs and CNVs. (**A**) CES1 rs1378942, (**B**) CHAT rs2177369, (**C**) COMT rs4680, (**D**) GSTM1 Indel CNV, **(E**) GSTP1 rs1138272 and rs1695, (**F**) GSTT1 Indel CNV, (**G**) NAT2 rs1041983, rs1208m, rs1700020, rs1799930, rs1799931m, rs1801279, and rs1801280, (**H**) SOD2 rs4880, (I) TPMT rs1142345, rs1800460, rs1800462, and rs1800584, (**J**) UGT1A1 rs35350960, rs14124874, rs4148323, and rs887829. Chi-square tests were used to compare the overall or global distribution of metabolizer geno-phenotypes between Groups A and B. The *p* values indicate those geno-phenotypes that are significantly different between patient groups A and B. Chi-square tests were used to compare the overall or global distribution of metabolizer geno-phenotypes between Groups A and B. The *p* values reference those geno-phenotypes that are significantly different between patient groups A and B.

**Figure 3 biology-12-01156-f003:**
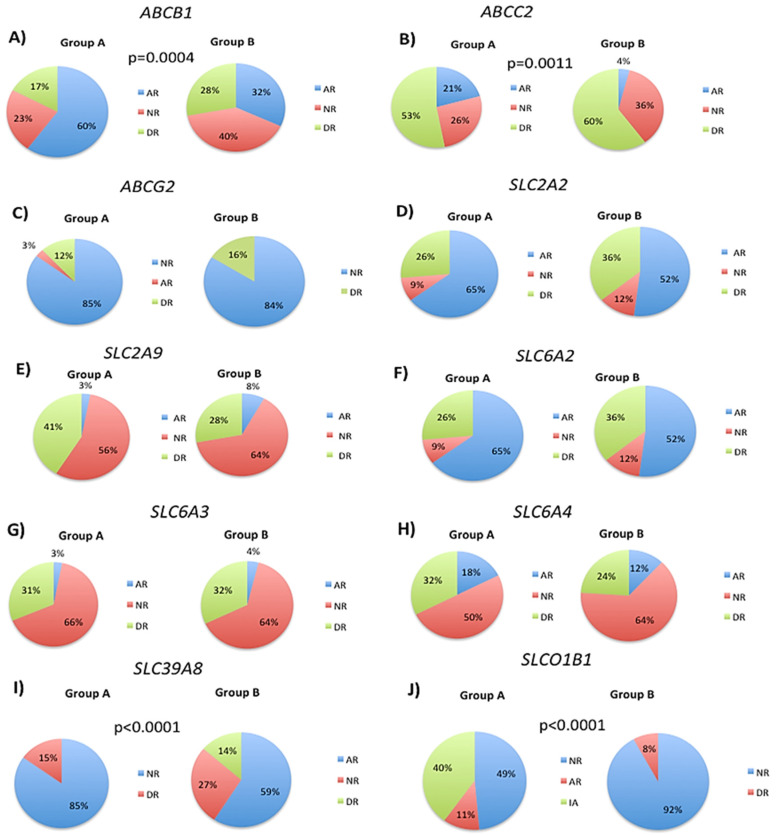
Pie charts showing the distribution of transport gene phenotypes in patient groups A and B. The frequencies of normal, deficient, and abnormal response phenotypes were determined for each gene, and specific SNPs were analyzed to investigate their effect on phenotypic responses to treatment. (**A**) ABCB1 rs1128503, rs1032582.01, rs2032582.02 and rs1045642, (**B**) ABCC2 rs717620, rs2273697, rs17222723 and rs3740066, (**C**) ABCG2 rs2231142, (**D**) SLC2A2 rs5400, (**E**) SLC2A9 rs16890979, (**F**) SLC6A2 rs5569, (**G**) SLC6A3 rs460000, (**H**) SLC6A4 rs2020936, (**I**) SLC39A8 rs13107325, (**J**) SLCO1B1 rs2306283, rs4149015, and rs1419056. Chi-square tests were used to compare the overall or global distribution of transport geno-phenotypes between Groups A and B. The *p* values indicate those geno-phenotypes that are significantly different between patient groups A and B.

**Figure 4 biology-12-01156-f004:**
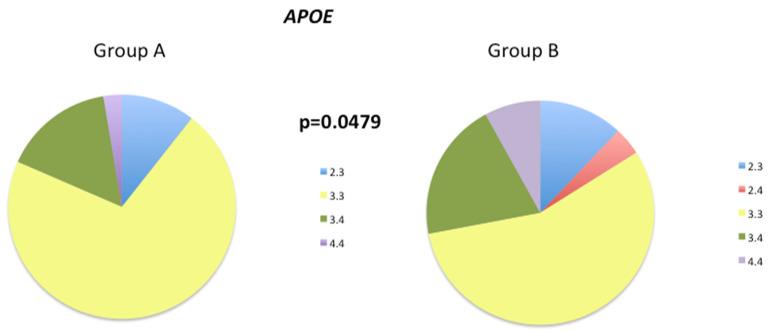
Distribution of *APOE* genotypes in Group A and Group B patients. Chi-square tests were used to compare the global distribution of *APOE* genotypes between Groups A and B; *p* < 0.05 indicates an overall significant difference for the *APOE* genotypes between these patient groups.

**Figure 5 biology-12-01156-f005:**
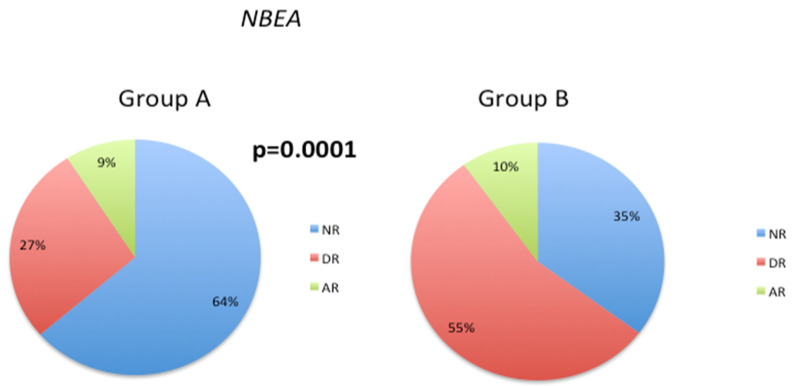
Frequency of different response phenotypes to acetylcholinesterase inhibitors (AChEIs) in Group A and Group B patients. Chi-square tests were used to compare the overall distribution of the *NBEA* response phenotype between Groups A and B; *p* < 0.05 indicates a significant difference for this response phenotype between these patient groups.

**Figure 6 biology-12-01156-f006:**
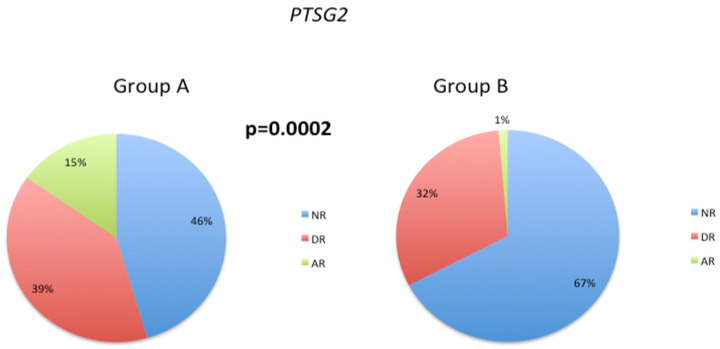
Distribution of *PTGS2* SNP rs5275 response phenotypes in Group A and Group B patients. Chi-square tests were used to compare the overall distribution of the *PTSG2* response phenotype between Groups A and B; *p* < 0.05 indicates a significant difference for the *PTSG2* response phenotype between these patient groups.

**Table 1 biology-12-01156-t001:** Summary of the pharmacogenetic genes that influence global DNA methylation levels during patient follow-up.

Metabolic (Phase I)	Metabolic (Phase II)	Transporter	Pathogenic
*CYP1A2*	*GSTP1*	*ABCB1*	*APOE*
*CYP2C9*	*NAT2*	*ABCC2*	*NBEA*
*CYP4F2*		*SLC2A9*	*PTSG2*
		*SLC39A8*	
		*SLCO1B1*	

## Data Availability

The data presented in this study are available on request from the corresponding author.

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
