# Peer review of "Influence of Metabolic, Transporter, and Pathogenic Genes on Pharmacogenetics and DNA Methylation in Neurological Disorders"

_biology, 2023, doi:10.3390/biology12091156_

Round 1
Reviewer 1 Report
This paper describes the impact of pharmacogenetics and DNA methylation on therapeutic outcomes and potential therapeutic targets for brain-related disorders. The study analyzed serum samples from two groups of patients: Group A, which experienced increased 5-methyl-cytosine (5mC) levels during clinical follow-up, and Group B, which did not exhibit any significant change in 5mC levels.
The researchers studied the relationship between global DNA methylation, drug-metabolizing enzymes, transport genes, and pathogenic gene phenotypes in these patient groups. Specific single-nucleotide polymorphisms (SNPs) in various metabolizing genes, such as CYP1A2, CYP2C9, CYP4F2, GSTP1, and NAT2, are claimed to be associated with differential drug responses.
SNPs in certain CYP genes are claimed to impact enzyme activity, leading to changes in phenotypic distribution between the two patient groups. The authors said that Group B has a lower frequency of normal metabolizers and a higher frequency of ultra-rapid metabolizers compared to patients in Group A. However, Group B did not show an improvement in 5mC levels during follow-up.
The researchers also observed significant differences in phenotype distribution between patient Groups A and B for several SNPs associated with transporter genes (ABCB1, ABCC2, SLC2A9, SLC39A8, and SLCO1B1) and pathogenic genes (APOE, NBEA, and PTGS2). These findings suggest that the interplay between pharmacogenomics (the study of how genetic variations influence drug response) and DNA methylation has important implications for improving treatment outcomes in patients with brain-related disorders.
However, this paper contains serious flaws.
1) The relevance to the scope of the journal. This is a technical paper on clinical medicine, not on fundamental biology. Therefore, it is suited much better for a more specialized medical journal.
2) What is more serious, despite this paper looks too technical for a high-profile journal, the authors did not apply any statistical analysis to any piece of their data! The paper contains a lot of numbers as the appearing results but they are senseless because of the absence of any traces of statistical analyses.
There is only one phrase containing the word “statistic*” in the whole paper
“Pharmacogenomic analysis of NAT2 phenotypes in response to AtreMorine show that the increase in DNA methylation is not statistically significant in patients with the fast acetylator phenotype [72].”
Do the authors realize that they report no statistically significant result in their own paper? Or did they just mechanically copy-paste this phrase from the cited paper, not understanding the meaning of “statistically significant“ piece?
Please provide extensive statistical analysis of the data
3) The reference list looks funny. I wonder, did the authors see it?
It would be good to polish English style
Author Response
REVIEWER 1
Comment 1:
The relevance to the scope of the journal. This is a technical paper on clinical medicine, not on fundamental biology. Therefore, it is suited much better for a more specialized medical journal.
Response:
Thank you to the reviewer. Our manuscript was a general submission to MDPI Biology. It was accepted and redirected by MDPI administrators to the Special Issue Epigenetic Modifications and Changes in Neurodegenerative Diseases and proceeded to peer review.
We would like to highlight that genetics and genomics are well-established subject areas within the field of Biology (https://www.mdpi.com/journal/biology/about). Moreover, MDPI Biology has previously published various papers on pharmacogenetics (https://www.mdpi.com/search?q=pharmacogenetics&journal=biology).
Comment 2:
What is more serious, despite this paper looks too technical for a high-profile journal, the authors did not apply any statistical analysis to any piece of their data! The paper contains a lot of numbers as the appearing results but they are senseless because of the absence of any traces of statistical analyses.
Response:
The authors apologize for this omission and thank the reviewer for pointing this out. In the revised manuscript, we have now applied and included statistical analysis of our data. This information has been added to each Figure, and has been described further in the Figure legend. We have also included a new section 2.6 (Statistical Analysis; page 5) under Materials and Methods.
Comment 3:
The reference list looks funny. I wonder, did the authors see it?
Response:
Thank you to the reviewer. The reference list has been corrected in the resubmitted manuscript.
Reviewer 2 Report
The study identified that SNPs in genes related to metabolic function, transporter, and pathogenic roles have potential to be involved in DNA methylation in neurological disorders, which may have an impact in pharmacogenetics of brain-related diseases.
Some revisions for clarification are needed to improve the manuscript.
1. The meaning of "multifactorial treatment" can be explained more in detail in line 46-47 in Introduction section.
2. The description on "Intel SNP in the GSTM1 gene" is unclear. Please clarify it in lines 254-257 in Results section.
3. The differences in "pharmacogenetic patient phenotypes" may be discussed in around lines 424-426 in Discussion section.
4. Table 1 may be revised to elaborate "the pharmacogenetic genes" in lines 559-561 in Discussion.
Some careful proofreading is needed.
Author Response
REVIEWER 2
Comment 1:
The meaning of "multifactorial treatment" can be explained more in detail in line 46-47 in Introduction section.
Response:
Thank you to the reviewer. We have now provided a more detailed explanation of "multifactorial treatment" in the second paragraph of the Introduction section of the revised manuscript.
In the context of our study, multifactorial treatment refers to a comprehensive approach aimed at addressing the complexities of neurodegenerative diseases, particularly Alzheimer's disease (AD). This approach involves tailoring treatments to meet the specific needs of each individual, considering the diverse factors contributing to the disease. To effectively target AD and its associated challenges, multifactorial treatments encompass a broad range of pharmacological strategies. These include the use of neuroprotective and anti-dementia drugs, medications targeting concurrent pathologies and neuropsychiatric disorders, and therapeutic interventions to address metabolic deficits.
The incorporation of pharmacogenetic procedures can further enhance the benefits of such treatments. Studies have shown that over 90% of AD patients may require multifactorial treatments, as they often present with coexisting conditions. For instance, hypertension, obesity, type 2 diabetes mellitus, hypercholesterolemia, hypertriglyceridemia, metabolic syndrome, hepatobiliary disorder, endocrine/metabolic disorders, cardiovascular disorder, cerebrovascular disorder, neuropsychiatric disorders, and cancer can be among these concurrent conditions. We therefore hope to offer a clearer understanding of the importance and scope of multifactorial treatment in managing neurodegenerative diseases.
Reference:
Cacabelos, R., Naidoo, V., Martínez-Iglesias, O., Corzo, L., Cacabelos, N., Pego, R., & Carril, J. C. (2022). Pharmacogenomics of Alzheimer's Disease: Novel Strategies for Drug Utilization and Development. Methods in molecular biology (Clifton, N.J.), 2547, 275–387.
Comment 2:
The description on "Intel SNP in the GSTM1 gene" is unclear. Please clarify it in lines 254-257 in Results section.
Response:
This description has been clarified in the resubmitted manuscript.
Comment 3:
The differences in "pharmacogenetic patient phenotypes" may be discussed in around lines 424-426 in Discussion section.
Response:
Thank you to the reviewer. We have now discussed the differences in pharmacogenetic patient phenotypes in the Discussion section of the revised manuscript.
Specifically, we observed a decrease in CYP1A2 and CYP2E1 normal metabolizers in Group B, highlighting the potential significance of CYP1A2 in response to DNA methylation changes. Similar trends were noticed for the CHAT gene, with a decrease in normal metabolizers from 35% to 24% and an increase in intermediate metabolizers from 27% to 48% in Group B.
Furthermore, our study revealed a higher percentage of patients with deficient responses for the ABCB1, SLC2A2, SLC6A2, and SLC39A8 genes in Group B. Notably, an increase in the presence of allele 4 of APOE was observed in Group B, suggesting a potential association with poor prognosis and pathogenicity in this cohort. Similar trends were also evident in the NBEA gene, with an increase in deficient response genotypes from 27% to 55%.
Comment 4:
Table 1 may be revised to elaborate "the pharmacogenetic genes" in lines 559-561 in Discussion.
Response:
The authors would like to request clarification of this comment by the reviewer as it was unclear. To the best of our knowledge, the information stated by the reviewer had been included in Table 1.
Reviewer 3 Report
This manuscript by Martinez-Iglesias and colleagues investigated the influence of metabolic, transporter, and pathogenic genes on pharmacogenetics and DNA methylation in patients with neurological disorders (ND). The content of this report is novel. I have some comments that need to be addressed first:
1. For the global DNA methylation (5mC), is there any gender, age or racial related differences in other studies? From this manuscript, 98 patients were recruited, but how many male and female patients were not mentioned in this study. If there were gender related differences in the global DNA methylation from other studies, the authors need to give a brief discussion.
2. For the SNP study, the authors used other methods such as QuantStudio system but not Sanger sequencing, it is well known that Sanger sequencing after PCR amplification of the targeted genes also can be used to determine the SNPs. The authors may add some discussion regarding why not using Sanger sequencing method.
3. Besides DNA methylation, there is also RNA methylation, I was wondering whether there is any change in RNA methylation in patients with ND. If there is relevant literature, please give a brief discussion.
Thank you!
Author Response
REVIEWER 3
Comment 1:
For the global DNA methylation (5mC), is there any gender, age or racial related differences in other studies? From this manuscript, 98 patients were recruited, but how many male and female patients were not mentioned in this study. If there were gender related differences in the global DNA methylation from other studies, the authors need to give a brief discussion.
Response:
Thank you to the reviewer for this feedback. We included a total of 98 patients in our study, with 53 male and 45 female participants. This information has now been added to both the Materials and Methods section, and the Discussion section of the manuscript.
With regard to the relationship between global DNA methylation levels and age, previous work by Salemeh et al. (2020) demonstrated a positive correlation between these two parameters. However, in our study, we found that this correlation was significant only in patients with Parkinson's disease (PD) and not in Group A patients. Therefore, while age may influence global DNA methylation levels in specific patient groups, it might not be a significant factor in Group A.
Concerning gender-related differences, we carefully examined the data and found no significant correlation between global DNA methylation and gender. The study population consisted of 53 male and 45 female participants, and the lack of significant differences in their DNA methylation levels suggests that gender may not play a significant role in this context. We have addressed these findings in the Discussion section, highlighting the observed correlations with age in PD patients and the lack of significant associations with gender in our study.
References:
Salameh, Y., Bejaoui, Y., & El Hajj, N. (2020). DNA Methylation Biomarkers in Aging and Age-Related Diseases. Frontiers in Genetics, 11, 171.
Martínez-Iglesias, O., Carrera, I., Carril, J. C., Fernández-Novoa, L., Cacabelos, N., & Cacabelos, R. (2020). DNA Methylation in Neurodegenerative and Cerebrovascular Disorders. International Journal of Molecular Sciences, 21(6), 2220.
Martínez-Iglesias, O., Naidoo, V., Corzo, L., Pego, R., Seoane, S., Rodríguez, S., Alcaraz, M., Muñiz, A., Cacabelos, N., & Cacabelos, R. (2023). DNA Methylation as a Biomarker for Monitoring Disease Outcome in Patients with Hypovitaminosis and Neurological Disorders. Genes, 14(2), 365.
Comment 2:
For the SNP study, the authors used other methods such as QuantStudio system but not Sanger sequencing, it is well known that Sanger sequencing after PCR amplification of the targeted genes also can be used to determine the SNPs. The authors may add some discussion regarding why not using Sanger sequencing method.
Response:
Thank you to the reviewer. In our study, we analyzed the pharmacogenetic profiles of patients who visited EuroEspes Biomedical Research Center. Our primary objective was to determine the sequence of the SNPs in our genes of interest. For this purpose, we chose to use the QuantStudio system instead of Sanger sequencing, as it facilitated a reliable and rapid analysis of the targeted genes.
We considered several key factors when making this decision:
First, the QuantStudio real-time PCR system provided customized arrays that were more effective in terms of time and reliability compared to Sanger sequencing for analyzing small regions of these genes.
Second, the high-throughput capabilities of the QuantStudio system allowed us to simultaneously analyze a large number of samples and SNPs, which was essential given the scope of our study involving numerous SNPs and a sizable sample set.
Third, the QuantStudio system demonstrated a high level of accuracy and sensitivity in SNP detection, ensuring the reliability of our results. While Sanger sequencing is indeed a robust and highly accurate method, its time-consuming and labor-intensive nature would have posed challenges, especially with the large number of samples and SNPs in our study.
Finally, Sanger sequencing requires a significant amount of DNA template for each reaction, which may not always be feasible based on the availability of samples.
Comment 3:
Besides DNA methylation, there is also RNA methylation, I was wondering whether there is any change in RNA methylation in patients with ND. If there is relevant literature, please give a brief discussion.
Response:
Thank you to the reviewer for this comment.
RNA methylation, particularly the N6-methyladenosine (m6A) modification, has emerged as a crucial regulatory mechanism in neurodegenerative diseases (NDs), shedding light on the intricate interplay among epigenetics, RNA metabolism, and neuronal function. The m6A RNA modification has been a focal point of research in neurons derived from induced pluripotent stem cells. These cells originate from patients with amyotrophic lateral sclerosis (ALS) or frontotemporal dementia (FTD), both of which are associated with a C9orf72 repeat expansion (Li et al. 2023).
In ALS and FTD, the m6A modification is closely linked to TDP43 binding and autoregulation (McMillan et al. 2023). The presence of extensive RNA hypermethylation in the ALS spinal cord corresponds to methylated TDP43 substrates. Furthermore, the canonical m6A reader, YTHDF2, has been identified as a modulator of TDP43-mediated toxicity, and its knockdown has shown promising results in extending the survival of neurons carrying ALS-associated mutations. In ALS and FTD, a reduction in the m6A RNA modification disrupts gene expression and accelerates neurodegeneration (Li et al. 2023). This highlights the potential for investigating the influence of pharmacogenetics on RNA methylation, providing valuable insights into the complex mechanisms underlying these diseases.
Abnormalities in m6A mRNA methylation have been linked to differentially expressed genes in Alzheimer's disease (AD) (Zhang et al. 2022). Dysregulation of mitochondrial and cytosolic tRNA m1A methylation has also been associated with age-related disorders such as AD. METTL3-dependent RNA m6A dysregulation contributes to neurodegeneration in AD through aberrant cell cycle events, emphasizing the complexity of epitranscriptomic control in the disease. This highlights the multifaceted role of RNA methylation in AD, underscoring its potential as a therapeutic target and its contribution to our understanding of neurodegenerative processes.
This information has been included in the last paragraph of the Discussion section.
References:
Li, Y., Dou, X., Liu, J., Xiao, Y., et al. (2023). Globally reduced N6-methyladenosine (m6A) in C9ORF72-ALS/FTD dysregulates RNA metabolism and contributes to neurodegeneration. Nature Neuroscience. DOI: 10.1038/s41593-023-01374-9.
McMillan, M., Gomez, N., Hsieh, C., Bekier, M., Li, X., Miguez, R., Tank, E. M. H., & Barmada, S. J. (2023). RNA methylation influences TDP43 binding and disease pathogenesis in models of amyotrophic lateral sclerosis and frontotemporal dementia. Molecular Cell, 83(2), 219–236.e7.
Zhang, R., Zhang, Y., Guo, F., Li, S., & Cui, H. (2022). RNA N6-Methyladenosine Modifications and Its Roles in Alzheimer's Disease. Frontiers in Cellular Neuroscience, 16, 820378.
Round 2
Reviewer 1 Report
The Authors addressed all my concerns.
The Article can be published.
A very small question about a phrase:
“For the Intel SNP in the GSTM1 gene, which is characterized by a complete deletion (GSTM1*0) …” (line 295).
SNP means “single nucleotide polymorphism”. It cannot be a complete deletion of a gene. May it be the loss-of-function polymorphism, not complete gene deletion?
And why is it called “Intel”? Google did not find “Intel SNP “ in relation to genes (except for preprint of the present paper). May it be “null polymorphism”?
English is fine.
Author Response
REVIEWER 1
Comment 1:
“For the Intel SNP in the GSTM1 gene, which is characterized by a complete deletion (GSTM1*0) …” (line 295). SNP means “single nucleotide polymorphism”. It cannot be a complete deletion of a gene. May it be the loss-of-function polymorphism, not complete gene deletion?
And why is it called “Intel”? Google did not find “Intel SNP “in relation to genes (except for preprint of the present paper). May it be “null polymorphism”?
Response:
We apologize to the reviewer for this error. We would like to clarify that both GSTT and GSTM polymorphisms are copy number variations and not SNPs. “Intel” has been corrected to “Indel”. This information has been corrected in the revised manuscript.
We hope that we have addressed the comments to the satisfaction of the reviewer.
Sincerely,
Dr Olaia Martínez-Iglesias
Department of Medical Epigenetics
EuroEspes Biomedical Research Center
Bergondo, 15165
Corunna, Spain
E-mail: epigenetica@euroespes.com